# Significant Roles of Notch *O*-Glycosylation in Cancer

**DOI:** 10.3390/molecules27061783

**Published:** 2022-03-09

**Authors:** Weiwei Wang, Tetsuya Okajima, Hideyuki Takeuchi

**Affiliations:** 1Department of Molecular Biochemistry, Nagoya University School of Medicine, 65 Tsurumai, Showa-ku, Nagoya 466-8550, Japan; oubibi82@yahoo.co.jp (W.W.); tokajima@med.nagoya-u.ac.jp (T.O.); 2Institute for Glyco-Core Research (iGCORE), Integrated Glyco-Biomedical Research Center, Nagoya University, Furo-cho, Chikusa-ku, Nagoya 464-8601, Japan; 3Department of Biochemistry, School of Pharmaceutical Sciences, University of Shizuoka, 52-1 Yada, Suruga-ku, Shizuoka 422-8526, Japan

**Keywords:** notch signaling, *O*-glycosylation, EGF repeat, cancer

## Abstract

Notch signaling, which was initially identified in *Drosophila* wing morphogenesis, plays pivotal roles in cell development and differentiation. Optimal Notch pathway activity is essential for normal development and dysregulation of Notch signaling leads to various human diseases, including many types of cancers. In hematopoietic cancers, such as T-cell acute lymphoblastic leukemia, Notch plays an oncogenic role, while in acute myeloid leukemia, it has a tumor-suppressive role. In solid tumors, such as hepatocellular carcinoma and medulloblastoma, Notch may have either an oncogenic or tumor-suppressive role, depending on the context. Aberrant expression of Notch receptors or ligands can alter the ligand-dependent Notch signaling and changes in trafficking can lead to ligand-independent signaling. Defects in any of the two signaling pathways can lead to tumorigenesis and tumor progression. Strikingly, *O*-glycosylation is one such process that modulates ligand–receptor binding and trafficking. Three types of *O*-linked modifications on the extracellular epidermal growth factor-like (EGF) repeats of Notch receptors are observed, namely *O*-glucosylation, *O*-fucosylation, and *O*-N-acetylglucosamine (GlcNAc) modifications. In addition, *O*-GalNAc mucin-type *O*-glycosylation outside the EGF repeats also appears to occur in Notch receptors. In this review, we first briefly summarize the basics of Notch signaling, describe the latest information on *O*-glycosylation of Notch receptors classified on a structural basis, and finally describe the regulation of Notch signaling by *O*-glycosylation in cancer.

## 1. Introduction to Notch Signaling

Notch signaling is a highly conserved cell signaling system in metazoans, as it controls multiple processes involved in the development of multicellular organisms, tissue homeostasis, and stem cell maintenance [1]. In humans, both gain- and loss-of-function mutations in the Notch pathway lead to numerous diseases, ranging from developmental syndromes to adult-onset disorders [2]. Therapeutic approaches to alter the activity of Notch signaling are subject to intense research and development [3]. Dysregulation of Notch signaling in many cellular contexts is also related to tumorigenesis, in which the Notch signaling pathway acts as a tumor suppressor, such as in small-cell lung cancer, and exhibits oncogenic functions, such as in T-cell acute lymphoblastic leukemia (T-ALL) [4].

Mammals possess four Notch receptors (NOTCH1–4) and five ligands, which consist of two types of canonical Delta/Serrate/LAG-2 (DSL) family ligands: three delta-like ligands (DLL1, DLL3, and DLL4), and two jagged ligands (JAG1 and JAG2) (Figure 1A) [5]. *Caenorhabditis elegans* has two Notch receptors (LIN12 and GLP1), while *Drosophila* has only one. In *Drosophila*, there are two canonical ligands: delta and serrate. *C. elegans* has four ligands: LAG-2, APX-1, ARG-1, and DSL-1 [6]. Notch receptors are type I transmembrane proteins. Similar to Notch receptors, canonical Notch ligands are also type-I transmembrane proteins, with tandem epidermal growth factor-like (EGF) repeats in their extracellular domains (ECDs), which facilitate their interaction with Notch receptors [6].

The functional Notch receptor is expressed on the cell surface as a non-covalently associated heterodimer consisting of ECD and the transmembrane domain followed by the intracellular domain (ICD) as the Notch ECD (NECD) is proteolytically cleaved by a furin-like convertase at site 1 (S1 cleavage) during trafficking through the Golgi apparatus (Figure 1B) [7].

The NECD contains 29–36 EGF repeats (36 in NOTCH1 and NOTCH2, 34 in NOTCH3, and 29 in NOTCH4) that are responsible for ligand-binding interactions [8,9]. These EGF repeats harbor sites for the addition of both *N*-linked and *O*-linked glycans [10,11,12]. Following the EGF repeats, there are three cysteine-rich LIN12-Notch repeats (LNRs) and a heterodimerization domain (HD) that functions as a negative regulatory region (NRR) that prevents ligand-independent activation of the Notch signaling pathway [13,14]. The ECD of DSL ligands contains a conserved *N*-terminal DSL domain and several EGF repeats. The ICD of Notch receptors (NICD) is a transcriptional co-activator that mediates transcriptional changes downstream of Notch activation. The NICD of *Drosophila* contains a RAM domain followed by six ankyrin (ANK) repeats that can bind to the CBF1/RBPjk/Su(H)/Lag1 (CSL) transcription factor [15,16], a transactivation domain (TAD), and a PEST (polypeptide enriched in proline (P), glutamic acid (E), serine (S), and threonine (T)) motif that regulates protein stability [17,18]. The TAD region is found in Notch1 and 2, but not in Notch3 and 4 in mammals [19].

After Notch receptors are translated, their ECDs undergo *O*-linked glycosylation initially in the endoplasmic reticulum (ER) and then in the Golgi apparatus, which is crucial for proper folding of the Notch receptor and its interaction with its DSL ligands (Delta, Serrate, and Lag-2). An individual EGF domain consists of a 30–40-amino acid sequence (Figure 2A). The EGF repeats are defined by the presence of six conserved cysteine residues forming three disulfide bonds that are essential for its three-dimensional structure. EGF repeats can be modified with *O*-glycans at distinct sites [9,10,11,12,20,21]. The three major types of *O*-linked glycosylation found on the EGF repeats of Notch pathway components are *O*-fucosylation, *O*-glucosylation, and *O*-*N*-acetylglucosamine (GlcNAc) modification, which exist as monosaccharides or extended forms.

Upon reaching the cell surface, the ECDs of Notch receptors from the signal-receiving cell interact with the trans-ligand (ligands from the neighboring signal-sending cell). This interaction is followed by endocytosis of the trans-ligand by the signal-sending cell, and endocytosis of the ligand on the signal-sending cell generates a pulling force that induces a conformational change in the NRR [25], exposing ADAM10/17 cleavage site 2 (S2), located on the extracellular side, about 12 amino acids away from the transmembrane domain, which is a key regulatory step in Notch activation [14], which allows ADAM-mediated processing (S2 cleavage) [26]. Following S2 cleavage, the γ-secretase complex containing presenilin and nicastrin mediates the intra-transmembrane cleavage at site 3 (S3 cleavage), releasing the NICD, which is subsequently translocated into the nucleus. In the nucleus, NICD interacts with the DNA-binding transcriptional factor CSL, which recruits the co-activator, mastermind-like (MAML), to form a ternary transcriptional complex that activates the transcription of downstream target genes [14,27]. During transcriptional activation, NICD is phosphorylated on its PEST domain and targeted for proteasome-mediated degradation by ubiquitin ligases, known as F-box and WD repeat domain-containing 7 (FBXW7). This limits the half-life of canonical Notch signaling [28].

## 2. Glycosylation in Notch Signaling

Protein glycosylation involves the addition of carbohydrate molecules to proteins. In general, protein glycosylation can occur both co-translationally or post-translationally and also contributes to the structural and functional characteristics of a large number of glycosylated proteins. It usually occurs in the ER and Golgi apparatus for secreted and transmembrane proteins and involves the addition of different forms of glycan to specific sites of proteins. The two major classes of protein glycosylation are *N*-linked and *O*-linked. *N*-Glycans are attached to an asparagine residue, whereas *O*-glycans are attached to serine or threonine residues. During animal development, glycosylation regulates various cell signaling pathways. The Notch signaling pathway is one of the most conserved cellular signaling pathways that plays different roles in animal development and tissue homeostasis [27,29].

Recent visualization of the structures of Notch receptors in complex with Notch ligands and those of Notch-modifying glycosyltransferases has helped us understand the significance of glycosylation in Notch regulation [27,30,31,32,33,34,35]. The foundation of Notch glycobiology was established in 2000, when fringe (Fng [36]), a key component of the Notch pathway, was shown to be a glycosyltransferase that can modulate the Notch–ligand interactions by adding GlcNAc to *O*-fucose on Notch EGF repeats [37,38]. Since then, studies have shown that NECDs can be modified with multiple types of glycans, and these modifications can be subdivided on the basis of *N*-glycans and *O*-glycans.

Three types of *O*-glycosylation on the EGF repeats of Notch receptors are *O*-glucosylation, *O*-fucosylation, and *O*-GlcNAc addition, that are initiated by *O*-glucose, *O*-fucose, and *O*-GlcNAc, respectively [39]. Notch signaling pathway activity can be influenced by post-translational modifications [14]. Each *O*-glycosylation reaction occurs at specific positions only in properly folded EGF repeats in Notch receptors or DSL ligands via specific glycosyltransferases. Most of these enzymes are responsible for these three types of *O*-glycosylations on EGF repeats of Notch receptors and their ligands are localized in the ER [40,41,42,43,44,45,46]. Following proper glycosylation in the ER and Golgi apparatus, Notch is transported to the cell surface. EGF repeats in the ECD of mouse Notch receptors have been widely analyzed by mass spectrometry, and most glycosylation sites have been thoroughly examined [12,22,23,24,39,47,48,49]. *O*-Fucosyltransferase-1 (OFUT1) functions as a chaperone to promote the folding and/or export of Notch to the plasma membrane [43,50].

*O*-Glycans play critical and multiple roles in the activation of Notch receptors, including protein folding/stability, trafficking, and ligand binding. Consequently, several human diseases are related to mutations in genes encoding the Notch-modifying glycosyltransferases.

### 2.1. O-Fucosylation

*O*-Fucose modification was first discovered after the isolation of amino acid fucosides from human urine in 1975 [51] and later found on the EGF repeats in NOTCH1 derived from Chinese hamster ovary (CHO) cells [52].

In *Drosophila* NOTCH, *O*-fucosylation occurs within the consensus sequence of EGF repeats with high stoichiometry [47]. POFUT1 (Ofut1 in *Drosophila*) in mammals, using GDP-fucose as a donor substrate, adds *O*-linked fucose to the EGF repeats with the consensus sequence in the ER [47,53,54]. *O*-Fucosylation occurs on specific serine or threonine residues within the consensus sequence, C2-X-X-X-X-(S/T)-C3 (where S/T is serine or threonine; C2 and C3 are the second and third cysteines of the EGF repeat, respectively; X can be any amino acid), on EGF repeats. A number of studies have revealed interesting aspects of *O*-fucose glycans on these EGF repeats in the regulation of Notch–ligand interactions [37,49,55].

The majority of the Ofut1/POFUT1 target sites in these proteins were efficiently modified, as confirmed by mass spectrometric analysis of *Drosophila* Notch and mouse NOTCH1 and NOTCH2 (Figure 2B, C) [22,24,47]. A recent study identified the X-ray crystal structure of mouse POFUT1 in compounds with different EGF domains, including Notch EGF12 and EGF26, and defined its EGF-domain binding properties [56]. The Blacklow group has reported the structures of human POFUT1 in free and GDP-fucose-bound states and assessed the effects of Dowling-Degos mutations on human POFUT1 function [57].

Genetic deletion or knockdown of *Ofut1* leads to the loss of Notch signaling, strongly suggesting that *O*-fucosylation plays a vital role in Notch signaling [58,59]. Similarly, POFUT1 is required for Notch signaling in mammals. Loss of *Pofut1* in mice is embryonically lethal due to a global loss of Notch signaling activity [60]. *O*-Fucose glycans directly participate in the binding of NOTCH1-DLL4 and NOTCH1-JAG1 and perform critical functions [22,31,32]. Furthermore, global loss of *O*-fucose on NOTCH reduces its binding to both Delta and Serrate ligands [55], and overexpression of *Ofut1* produced soluble Notch fragments that significantly enhances the binding of Notch to Serrate [55]. Studies have suggested that Ofut1 plays non-enzymatic roles as a chaperone in Notch protein folding and its exit from the ER in *Drosophila* [43] as well as in endocytic trafficking of Notch [50]. Mutations in individual *O*-fucosylation sites of *Drosophila* Notch (EGF8, EGF9, and EGF12) and their pairwise combinations can strongly influence Notch signaling, but do not reduce the cell surface expression of Notch [61].

*O*-Fucose monosaccharides can be elongated with GlcNAc residues using Fng. Fng, a secreted protein that was originally named due to mutants exhibiting tissue loss at the edge or “fringe” of fly wings, was initially identified as a critical regulator in *Drosophila* wing development, which is a phenotype similar to *Notch* [36]. Further investigations have demonstrated that this elongation by Fng modulates the Notch activity in *Drosophila* [62]. *FNG* inhibits the Notch response to *Serrate (SER)* expressed in dorsal cells but enhances the Notch response to *Delta* expressed in ventral cells [62]. Initial observations showed that secreted FNG inhibited SER by interacting with NOTCH [63]. However, in 2000, two studies revealed that Fng is a glycosyltransferase that catalyzes the addition of GlcNAc to *O*-fucose on Notch EGF repeats in a β1,3-linkage [37,38]. Mammals have three homologs of *Fng*, including manic Fng (MFng), radical Fng (RFng), and lunatic Fng (LFng), whereas *Drosophila* has only one Fng protein [36,64,65]. The discovery of the Fng glycosyltransferase activity in 2000 suggested an association between the Notch signaling pathway and glycobiology.

Furthermore, β1,4-linked galactose and α2,3- or α2,6-linked sialic acid are added to the GlcNAc-fucose-*O* disaccharide on EGF repeats of mammalian proteins, to form tri- and tetrasaccharides successively [10]. It should depend on cell type which galactosyltransferases or sialyltransferases are responsible for those extensions. Whether the GlcNAc-fucose-O disaccharides can be extended to tri- or tetra-saccharides has not been reported in flies [10,66]. Overall, *O*-fucosylation modification is an important factor in the regulation of Notch signaling pathways.

### 2.2. O-Glucosylation

In 1988, *O*-glucose was found to be attached to EGF repeats of the bovine blood coagulation factors VII and IX [67]. In *O*-glucosylation, *O*-glucose glycans are added to the serine residue between the first and second cysteine residue within the consensus sequence, C1-X-S-X-(P/A)-C2 (where the modified serine is underlined, X represents any amino acid, and C1 and C2 are the first and second cysteine residues of the EGF repeat, respectively), of EGF repeats by POGLUT1 (rumi in *Drosophila*) based on the strict recognition of the amino acid sequences and the folding status of EGF repeats, the Rini group also reported the structure of human POGLUT1 in complexes with three different EGF-like domains and either UDP, a donor substrate analog, or a slow substrate [30,34,40,44].

This gene was first discovered in human hematopoietic stem/progenitor cells isolated from a patient with myelodysplastic syndrome who had undergone leukemic transformation, and referred to the similarity of its major recognizable domain with the CAP10 protein of *Cryptococcus neoformans*. This gene was named as human CAP10-like protein 46 kDa (*hCLP46*) [68]. *hCLP46* and its mouse homolog (formerly called *KTELC1*) were renamed as *POGLUT1/Rumi* after the discovery of the enzymatic activity of *Drosophila* Rumi and its mammalian homologs [40,48,69]. Moreover, POGLUT1/Rumi can add an *O*-xylose residue as well as an *O*-glucose residue to the same consensus sequence with a diserine motif [30,48].

Unlike POFUT1, POGLUTs and Rumi cannot add *O*-glucose to threonine residues, but can only glucosylate serine residues within the consensus sequence, which may be because POGLUT1 has Asp133, which is a catalytic base that can activate the hydroxyl group of serine, not threonine, by following an SN2 inversion mechanism [30,70] When replacing serine, threonine fails to adopt the optimal posture for nucleophilic activation due to steric hindrance [30], underlying the preference for serine over threonine for *O*-glucosylation. Similar to POFUT1, POGLUT1 is also localized to the ER and modifies EGF repeats that are properly folded [44,48]. POGLUT1 is also a soluble protein that has a lysine-aspartic acid-glutamic acid-leucine motif that can prevent secretion from the ER.

Recently, we reported that two mammalian homologs of POGLUT1/Rumi (KDELC1 and KDELC2) are unable to add *O*-glucose to EGF repeats harboring the POGLUT1 consensus sequence [41,48], but can add *O*-glucose to a specific serine residue in a motif between the third and fourth cysteine residues in EGF repeats [41]. It was revealed that *O*-hexose is attached to the serine residue between the third and fourth cysteine residues by the analysis of mass spectrometry and X-ray crystallography of the 11th EGF repeat of the NOTCH1 receptor [31,71]. This serine residue did not correspond to the site modified by POGLUT1. Subsequently, we showed that *O*-hexose is specifically *O*-glucose. Similar to POGLUT1 and POFUT1, these two enzymes, which were renamed POGLUT2 and POGLUT3, respectively, are protein *O*-glucosyltransferases that add *O*-glucose to the specific serine residue of EGF11 of NOTCH1 and EGF10 of NOTCH3 in the sequence, C3-X-N-T-X-G-S-F-X-C4, and modify only properly folded EGF repeats [41]. A catalytic CAP10 domain and a KDEL-like localization signal exist on POGLUTs/rumi-encoding glucosyltransferases. The CAP10 protein was first discovered in *C. neoformans* [72], and most proteins with the CAP10 domain also have an ER retention signal. According to the results of the Basic Local Align Search Tool (BLAST) analysis, POGLUT1 shares a higher level of identity (52%) with Rumi than POGLUT2 (37%) and POGLUT3 (38%), but POGLUT2 and POGLUT3 have an additional filamin-like domain. When all three human POGLUTs were expressed in flies, only POGLUT1 could rescue the Rumi loss-of-function phenotype [48], while POGLUT2 and POGLUT3 were unable to do the same. However, to date, POGLUT2/3 has shown a distinct preference for EGF acceptor substrates to only glucosylate the specific sites on one EGF repeat in human NOTCH1 and NOTCH 3 (EGF11 and EGF10, respectively) but not human NOTCH2 [41]. Knockout studies in model organisms and human disease-linked mutations remain to be evaluated. A recent study from the Haltiwanger lab showed that POGLUT2 and POGLUT3 preferentially modify fibrillins and latent TGF-β-binding protein 1 (LTBP1) [73]. In contrast to POGLUT1-mediated *O*-glucosylation, no evidence of an elongated form of this POGLUT2/3-mediated *O*-glucosylation has been reported so far.

In mammalian cells, *O*-glucose is essential for Notch signaling activity, whereas the binding between Notch receptors and ligands is not influenced by the elimination of POGLUT1 [69]. Loss of *rumi* and mutations in the *O*-glucosylation sites of *Drosophila* Notch lead to a temperature-sensitive loss of Notch signaling by affecting the ADAM-dependent cleavage [40,69,74,75]. However, no reduction in the surface expression of the Notch receptor was found in imaginal disks harboring the *rumi* mutant clones or expressing Notch with mutations in its *O*-glucosylation sites [40,74].

*O*-Glucose monosaccharides added by POGLUT1 to EGF repeats are attached to serine residues, and can be further extended to xylose-glucose-*O* disaccharides and xylose-xylose-glucose-*O* trisaccharides by GXYLT and XXYLT1 [76,77,78]. Mammals have two GXYLT enzymes, GXYLT1 and 2, while *Drosophila* has only one GXYLT, called *Shams* [77,79]. Moreover, the xyloside xylosyltransferase enzyme is XXYLT1 in mammals and Xxylt in *Drosophila* [76,80]. In mouse NOTCH1 and NOTCH2, all predicted POGLUT1 target sites were *O*-glucosylated, and most of them could be extended to a trisaccharide form (Figure 2B,C) [12,39], but only a subset of *O*-glucosylated EGF repeats of Notch is xylosylated in *Drosophila* [47,79]. Similar to POGLUT1 and POFUT1, these three mammalian xylosyltransferases also prefer to utilize properly folded *O*-glucosylated EGF repeats as acceptor substrates [44].

The extension of xylose to *O*-glucose monosaccharides aids in the normal function of Notch in *Drosophila* [79,80,81]. The first xylose of *O*-glucose prevents Notch from trafficking to the cell surface [12,79,82,83] and destabilizes the EGF repeats, and the addition of a second xylose restabilizes the EGF repeats after they are destabilized [84]. The modification of xylose residues by *Shams* on a subset of EGF repeats of Notch can decrease *Drosophila* Notch signaling in specific contexts [79].

Moreover, the loss of *Shams* results in the loss of both xylose residues from the Notch EGF repeats, and in *Shams* mutant flies, the loss of the second xylose contributes to the strengthening of Notch signaling activity. The overexpression of human *GXYLT1* leads to a decrease in Notch signaling in the *Drosophila* wing [79]. Furthermore, the amplification of *XXYLT1* has been discovered in various cancer types in which Notch signaling activity is inhibited [35,85]. These observations suggest that the addition of a second xylose to Notch EGF repeats by XXYLT1 might inhibit the activation of Notch signaling. Biochemical and genetic analyses of the contribution of xylose residues to Notch signaling in *Drosophila* indicated that the first xylose extension can regulate Delta-mediated Notch signaling, while the second xylose calibrates it only in sensitized genetic backgrounds [80]. Although it has been implied that POGLUT1 is involved in the proper transport of the Notch receptor and quality control in the ER, the exact function of this modification in the regulation of the Notch pathway has not yet been well-established.

### 2.3. O-GlcNAcylation

In 2008, the Okajima group originally discovered the *O*-GlcNAc modification on *Drosophila* Notch EGF20 by detecting *O*-glycans on the ECD of *Drosophila* Notch using an antibody that recognizes *O*-GlcNAc (CTD110.6) [86]. Later, *O*-GlcNAc modifications of the EGF repeats in *Drosophila* Delta, Serrate, and Dumpy were subsequently discovered [86,87]. *O*-GlcNAc is added to a serine or threonine residue within the putative consensus sequence of *O*-GlcNAc, C^5^-X-X-G-X-(T/S)-G-X-X-C^6^, based on experimental mapping in *Drosophila* Notch and mouse NOTCH1 by mass spectrometry (Figure 2B) [23,47,88,89].

The enzyme responsible for extracellular *O*-GlcNAcylation of *Drosophila* EGF repeats has been identified and designated as EGF domain-specific O-GlcNAc transferase (EOGT) in *Drosophila* by the Okajima group [87] and EOGT1 in mammals [90]. Similar to POFUT1 and POGLUT1-3, EOGT is also localized to the ER and modifies properly folded EGF repeats [87].

Despite the presence of *O*-GlcNAc modification consensus sites in 18 EGF repeats of *Drosophila* Notch, only five sites with robust *O*-GlcNAc modifications were found in the Notch derived from *Drosophila* Schneider 2 (S2) cells and embryos [47]. In contrast, mouse NOTCH1 possesses 17 EGF repeats with *O*-GlcNAc consensus sites, most of which are modified by *O*-GlcNAc [91].

Recently, mutations in *EOGT* were detected in patients with autosomal recessive Adams–Oliver syndrome (AOS), a rare congenital disorder characterized by aplasia cutis congenita and terminal transverse limb defects [92,93,94,95]. There were no obvious defects in Notch signaling in flies with loss of *EOGT* [87], whereas genetic interactions between *EOGT* and Notch pathway components have been reported [96]. The regulation of Notch signaling by O-GlcNAc glycans in endothelial cells is required for optimal vascular development [97]. Decreased Notch binding to DLL1 or DLL4 was detected in *EOGT*-deficient cells, whereas Notch binding to JAG1 was not influenced by the loss of *EOGT* [97]. In mammals, *O*-GlcNAc can be further elongated to a Gal-GlcNAc-*O* disaccharide and a sialic acid (Neu5Ac)-Gal-GlcNAc-*O* trisaccharide by galactose and sialic acid, respectively [23,90], although this extension appears to be found only in a sub-class of *O*-GlcNAcylated EGF repeats of the mouse NOTCH1 [91]. Like the extension of *O*-fucose disaccharides mentioned above, the linkage(s) and responsible glycosyltransferase(s) may vary depending on cell type which expresses Notch proteins. Moreover, the biological significance of this elongation of *O*-GlcNAc glycans remains to be elucidated.

## 3. Significance of Notch *O*-Glycosylation in Cancer

The relationship between Notch signaling and cancer has been well-studied and summarized in various reviews [98,99,100,101,102]. Notch signaling is frequently altered in T-ALL [103], CLL [104,105], diffused large B-cell lymphoma [106,107], mantle cell lymphoma [108], gastric and esophageal cancer, colorectal cancer, uterine corpus endometrial cancer [109], breast cancer [110,111,112], and non-small-cell lung cancer (NSCLC) [113]. Notably, different types of cancers exhibit aberrant expression of Notch-modifying glycosyltransferase genes, such as *POFUT1*, *POGLUT1*, or *Fng* (Table 1) [114]. Increased expression of *POFUT1* and *POGLUT1* has been found in several cancers, including brain tumors, hepatocellular carcinoma, colorectal cancer, and oral squamous cell carcinoma [115,116,117,118]. Since Notch signaling plays an important role in hematological cancers and *O*-glycosylation is an important regulator of Notch signaling, it is highly likely that *O*-glycosylation plays an important role in hematological cancers; however, this remains to be clarified. On the other hand, several important reports have been made in solid tumors, and we will introduce the relevant reports for each type of *O*-glycan structure.

### 3.1. O-Fucosylation of Notch Receptors in Cancer

Among the types of *O*-linked glycosylation of Notch receptors, *O*-fucose glycosylation was the first to be discovered as a regulator of Notch signaling; thus, this type of *O*-glycosylation was also the first to be studied for its importance in cancer. Higher expression of *POFUT1* in gliomas compared to normal cells has been reported [115]. *POFUT1* is localized in the 20q11.21 region. The 20q11.21 chromosomal region is frequently amplified in tumor cells such as in hepatocellular carcinoma (HCC) [117], breast cancer [119], gastric cancer [120], acute myeloid leukemia [121] and colorectal cancer (CRC) with poor prognosis [121,122]. In CRC, *POFUT1* expression and the copy number of the 20q11-13 amplicon are positively correlated [116]. These studies suggest that *POFUT1* plays a significant role in cancer development.

In stage I CRC, overexpression of *P**OFUT1* was found, and high expression of *POFUT1* was associated with the metastatic process [123]. In gastric cancer, increased *POFUT1* expression is associated with some clinical features, such as higher tumor-node-metastasis (TNM) staging and tumor differentiation states, indicating that *POFUT1* might act as a potential biomarker in gastric cancer [124]. In esophageal cancer stem-like cells (CSLCs), *POFUT1* and *2* are upregulated compared to adherent cells [125]. In contrast, low *POFUT1* mRNA expression is associated with a higher risk of overall and cancer-specific death in muscle-invasive bladder cancer (MIBC) treated with radical cystectomy. MIBC patients with decreased *POFUT1* mRNA levels showed poor outcomes for overall survival, cancer-specific survival, and disease-free survival [126].

Glycosyltransferases that modify Notch receptors, including POFUT1, modify proteins that contain EGF repeats with a consensus sequence for each *O*-glycosylation, so the effect of POFUT1 on cancer cell behavior is not necessarily mediated by Notch signaling. The crosstalk between *POFUT1* and Notch signaling in cancer has been previously described. In breast cancer, overexpression of *POFUT1* and activated NOTCH1 signaling was associated with lymph node metastasis and advanced tumor stage, leading to a poor prognosis [127]. In hepatocellular carcinoma cells, *POFUT1* overexpression promoted the binding of the Notch ligand DLL1 to the Notch receptor, and then activated the Notch signaling pathway, indicating that an aberrantly activated POFUT1-Notch pathway is involved in HCC progression [128]. Furthermore, Cav-1 can upregulate the expression of *Pofut1*, which activates the Notch pathway. Cav-1 can enhance invasion and metastasis by upregulating *Pofut1* expression in mouse HCC in vitro and in vivo [129]. In CRC, *POFUT1* silencing inhibited cell proliferation, decreased cell invasion and migration, arrested cell cycle progression, and stimulated CRC cell apoptosis in vitro and suppressed CRC tumor growth and transplantation in vivo [130]. Furthermore, structure and function studies on seven missense mutations in human *POFUT1* in rare CRC cases showed that six of the missense mutations lead to an increase in the protein *O*-fucosyltransferase activity in vitro [131]. High levels of *POFUT1* were found in glioblastoma (GBM) tissue, and GBM patients with high *POFUT1* expression had a shorter survival rate. Overexpression of *POFUT1* enhanced the proliferation and invasion of GBM cells, whereas inhibition of *POFUT1* significantly reduced the proliferation and invasion of GBM cells. This may be because *POFUT1* silencing inhibits Notch signal activation, resulting in reduced expression of *HES1* and *HEY1*, which indicated that *POFUT1* acts as a tumor promoter in GBM by enhancing the activation of Notch signaling [132]. Thus, it has been reported that in various cancers, the expression levels of POFUT1 alter Notch signaling in cancer cells, resulting in altered malignant behavior of cancer cells. However, the molecular mechanisms underlying this need to be further investigated. It is not clear whether the altered expression of *POFUT1* alters the levels of *O*-fucosylation in Notch receptors. In addition to its glycosyltransferase activity, POFUT1 has been shown to have a chaperon function for the Notch receptor, at least in the *Drosophila* system. From the viewpoint of glycobiology, it is still necessary to analyze how POFUT1 regulates Notch signaling in cancer cells.

In addition to alterations in *POFUT1* expression levels, *O*-fucose site mutations have been detected in anaplastic large cell lymphoma, a type of T-cell lymphoma [133]. The two mutations, p.T311P and p.T349P, of NOTCH1 result in the loss of the *O*-fucose site in the eighth and ninth EGF repeats in the ECD of NOTCH1, respectively. When the mutants and wild-type NOCTH1 were expressed in HEK293T cells, the mutants caused an increase in cell proliferation and the transcriptional activation of *HES1* and *HEY1*, which are downstream target genes of Notch signaling. It is still unclear whether these effects were due to the loss of *O*-fucosylation at specific EGF repeats of NOTCH1 and/or the mutation itself. Notably, it was shown that *O*-fucosylation on the ninth EGF repeat of NOTCH1 is important for its trafficking [24].

LFNG, which is a GlcNAc-transferase acting on *O*-fucose, has been well-studied in the context of cancers. Breast cancers and their association with Notch signaling have been well-studied [134]. Loss of *LFNG* can elevate Met/insulin-like growth factor 1 receptor (Igf-1R) signaling, which contributes to basal-like breast cancer (BLBC) (Figure 3) [135]. In this model, BLBC cells showed JAG1-induced Notch hyperactivation [135]. Recently, the Haltiwanger group, using a combination of mass spectrometry and cell-based reporter assays, has shown that LFNG-dependent GlcNAc-elongation of *O*-fucose glycans promotes DLL-induced activation of NOTCH1 and NOTCH2, while inhibiting JAG1-induced activation of NOTCH1 and NOTCH2 to varying degrees [22,24]. It is highly likely that LFNG acts in a manner similar in BLBC. Continuous activation of Notch signaling leads to amplification of the chromosome 7q31 locus, which contains the Met and Caveolin genes [135]. Human BLBC also shows low *LFNG* expression, increased MET signaling, and CAVEOLIN accumulation. For patients with triple-negative breast cancer with MET overexpression and Notch hyperactivation, the combined targeting of these two pathways may offer a new therapeutic strategy [136]. Claudin-low breast cancer (CLBC) is a disease with a poor prognosis and is biologically characterized by stemness and mesenchymal features. LFNG and p53 cooperatively suppress mesenchymal stem-like breast cancer, which is a poor-prognosis molecular subtype with stemness and mesenchymal features [137]. LFNG has also been reported to have a potent tumor-suppressive function in KRAS-mediated pancreatic cancer [138]. *LFNG* expression can also regulate the metastasis of melanoma as the ability to metastasize weakly metastatic melanoma cells was strongly enhanced in vivo in the condition of loss of *LFNG*, and the phenotype could be rescued with the *LFNG* cDNA in mouse melanoma cell lines [114].

Manic Fringe (Mfng) is highly expressed in CLBC and functions as an oncogene. The silencing of *MFNG* in CLBC cell lines reduces cell migration, tumor globule formation, and tumorigenicity in vivo, which is associated with a reduction in the number of stem cell-like cells [139]. MFNG inhibition of the JAG1-dependent Notch signaling pathway has been shown to be key to inhibiting the development of a subset of CRC. The 5-year survival rate of CRC patients with high MFNG levels was significantly higher than that of patients with low MFNG levels [140]. Recently, Cheng and co-workers showed that MFNG proteins were expressed in the nuclear and cytoplasmic compartments of normal kidney and renal cell carcinoma (RCC), although the expression of MFNG proteins did not show a significant association with clinical parameters of RCC patients. They assumed that MFNG could be a potential therapeutic molecular marker for RCC since they found that the association between CD20^+^ B cells and epithelial MFNG had statistically borderline insignificance in the limited cohort [141]. Moreover, they also found the role of MFNG in angiogenesis. They showed that MFNG was expressed in the endothelial cells and that the elevated expression of MFNG was also found in clear cell renal cell carcinoma (ccRCC). *MFNG* knockdown in endothelial cells decreased cell viability and migration. Moreover, ccRCC cell motility during co-culture of ccRCC cell cells with endothelial cells was reduced by knockdown of *MFNG* in endothelial cells [142]. Thus, these studies warrant further investigation into the roles of MFNG in the different types of cancers.

### 3.2. O-Glucosylation of Notch Receptors in Cancer

Information on the relationship between *O*-glucose glycosylation and cancer is much more limited than that on *O*-fucosylation, and research is lagging behind. Although its molecular function is unknown, the gene encoding POGLUT1 was first cloned from CD34^+^ cells derived from patients with myelodysplastic syndrome [68]. The authors named it human CAP10-like protein 46 kDa (hCLP46) and showed that overexpression in U937 cells enhanced proliferation. A subsequent study showed that upregulation of *POGLUT1* expression was also detected in hematopoietic malignancies, including primary acute myelogenous leukemia and T-ALL [143]. In addition, *POGLUT1* is amplified and overexpressed in NSCLC [144]. RNAi-mediated *POGLUT1* knockdown in A549 NSCLC cell lines caused a significant reduction in the expression of *HEY1* and *HES2*, both of which are Notch downstream target genes and inhibit cell proliferation, migration, and survival. It would be beneficial to further clarify the molecular mechanism by which POGLUT1 modulates Notch receptor glycosylation and thereby regulates Notch signaling in this specific context. Nonetheless, POGLUT1 is a novel negative prognostic factor and a potential therapeutic target for NSCLC.

While much remains unclear, the importance of xylosyl-extension of *O*-glucose glycans may be worth considering in the type of cancer where Notch signaling acts in a tumor-suppressive manner. Compared to healthy subjects, the expression of *GXYLT2* is significantly upregulated in patients with AML [145]. Although the functional significance of these findings needs to be investigated, the Tohda group reported that the protein expression of GXYLT1, GXYLT2, and XXYLT1 is increased upon JAG1- or DLL1-stimulation in AML cell lines [146]. GXYLT2 expression levels were significantly increased in gastric cancer tissues and were significantly correlated with poor survival [147]. The amplification of *XXYLT1* has been detected in many types of cancers, such as lung, esophagus, and head-and-neck-derived SCC, in which Notch signaling may have a tumor-suppressive role [35]. In esophageal cancer, patients with *XXYLT1* amplification have a poor prognosis compared to those without *XXYLT1* amplification [148]. In *Drosophila*, the activation of Notch signaling is downregulated after xylosyl-extension of *O*-glucose glycans [79,80,81]. The relationship between xylosyl-extension and Notch signaling in mammals remains elusive. If xylosyl-extension of *O*-glucose glycans negatively regulates mammalian Notch signaling, the treatment that targets Notch-modifying xylosyltransferases may provide a good perspective for specific types of cancer treatment strategies.

### 3.3. O-GlcNAcylation and O-GalNAcylation of Notch Receptors in Cancer

The abnormal expression of Notch receptors, ligands, and their downstream target genes in pancreatic ductal adenocarcinoma (PDAC) suggests that Notch signaling plays a role in the development and progression of pancreatic tumors [149,150,151]. *O*-GlcNAcylation also plays an important role in PDAC development [152]. The expression levels of EOGT and Shc SH2-binding protein 1 (SHCBP1) were significantly increased in patients with PDAC, which was associated with a worse prognosis. Overexpression of SHCBP1 promotes the proliferation, migration, and invasion of pancreatic cancer cells, while inhibition of SHCBP1 and EOGT inhibits these malignant processes in vitro. Researchers have further elucidated the molecular mechanisms by which EOGT and SHCBP1 enhance the *O*-GlcNAcylation of NOTCH1, thus promoting the nuclear localization of NICD and inhibiting the transcription of E-cadherin and P21 in pancreatic cancer cells [153].

The identification of *N*-acetylgalactosaminytransferase 11 (*GALNT11*) as a novel molecular marker in Notch-mediated chronic lymphocytic leukemia (CLL) provides a new perspective for understanding Notch glycosylation [154]. In B-cell chronic lymphocytic leukemia (B-CLL), NOTCH1, NOTCH2, and their ligands (JAG1 and JAG2) are constitutively expressed. The hyperactivity of NOTCH1, NOTCH2, JAG1, and JAG2 in malignant B cells can help malignant B cells become resistant to apoptosis [155]. *GALNT11* was identified as a heterotaxy gene that transfers a GalNAc residue to a specific threonine residue near the S2 cleavage site, which promotes ADAM-mediated S2 processing of NOTCH1 [156]. Further mechanistic investigation of mucin-type *O*-glycosylation outside the EGF repeats in NOTCH1 by GALNT11 in the context of CLL would be worthwhile.

## 4. Concluding Remarks

In this review, we described the glycosylation of the Notch receptor, which plays a central role in the Notch signaling pathway. In particular, genetic, biochemical, and cell biological analyses of *O*-linked glycans in model organisms and human pathologies have clearly shown that they are important regulators of Notch receptor activation. The development of tools for in vivo monitoring of Notch receptor glycosylation will greatly facilitate the study of glycosylation functions.

Research on the importance of Notch receptor glycosylation in cancer has just begun, and more information will be accumulated in the future. We, and hopefully our readers, believe that this will lead to the development of efficient technologies for the diagnosis, treatment, and prevention of cancer on the basis of glycosylation. Moreover, the combination of these technologies with existing ones will further contribute to the advancement of personalized medicine.

## Figures and Tables

**Figure 1 molecules-27-01783-f001:**
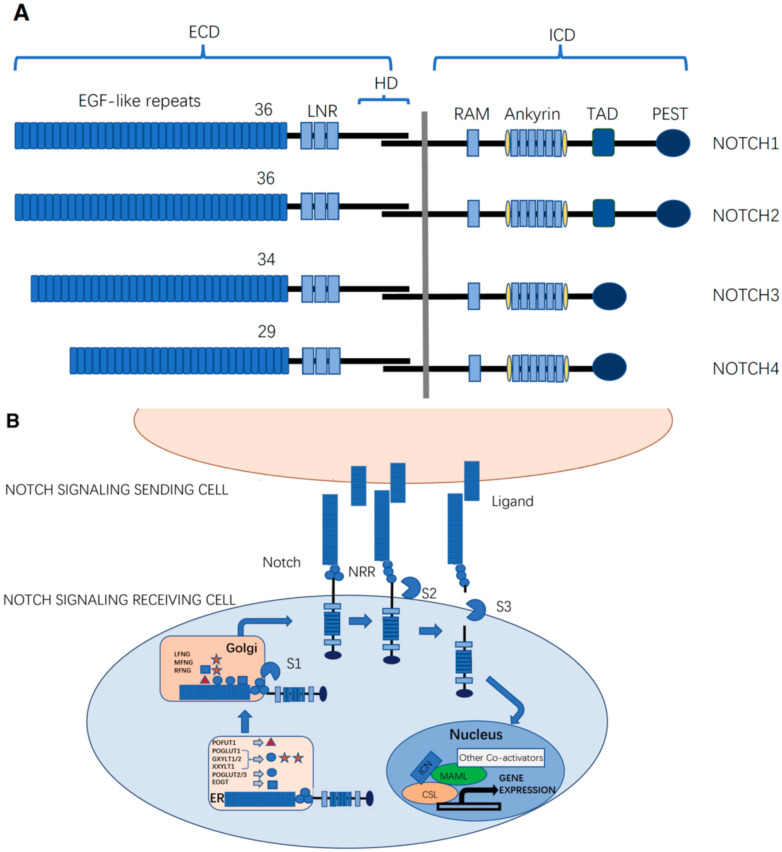
Notch receptors and components of the Notch signaling pathway. (**A**) Molecular structures of Notch receptors. Mammals possess four Notch receptors (NOTCH1–4), The Notch extracellular domain (NECD) contains 29–36 epidermal growth factor-like (EGF) repeats (36 in NOTCH1 and NOTCH2, 34 in NOTCH3, and 29 in NOTCH4) that are responsible for ligand binding. Following the EGF repeats, there are three cysteine-rich LIN12-Notch repeats (LNRs) and one heterodimerization domain (HD) which functions as a negative regulatory region (NRR) that prevents ligand-independent activation of the Notch signaling pathway. The TAD region is found in Notch1 and Notch2, but not in Notch3 and Notch4 in mammals. (**B**) Notch signaling pathway and its activation. Following proper glycosylation in the endoplasmic reticulum (ER) and Golgi apparatus, NECD is proteolytically cleaved by a furin-like convertase at site 1 (S1 cleavage), and the functional Notch receptor is expressed on the cell surface as a non-covalently associated heterodimer, consisting of the extracellular domain (ECD) and transmembrane domain followed by the intracellular domain (ICD). Upon reaching the cell surface, the ECDs of Notch receptors from the signal-receiving cell interact with the ligands from the neighboring signal-sending cell. This interaction is followed by the endocytosis of the trans-ligand on the signal-sending cell. The endocytosis of the ligand on the signal-sending cell generates a pulling force that induces a conformational change in the NRR, exposing the S2 cleavage site. A disintegrin and metalloproteinase (ADAM) proteases cleave the S2 site of Notch receptors, followed rapidly by γ-secretase-dependent S3 cleavage, releasing the ICD of Notch receptors (NICD), which is subsequently translocated into the nucleus. In the nucleus, the NICD interacts with the DNA-binding transcriptional factor CBF1/RBPjk/Su(H)/Lag1 (CSL), which recruits the co-activator, mastermind-like (MAML), to form a ternary transcriptional complex to activate the transcription of downstream target genes.

**Figure 2 molecules-27-01783-f002:**
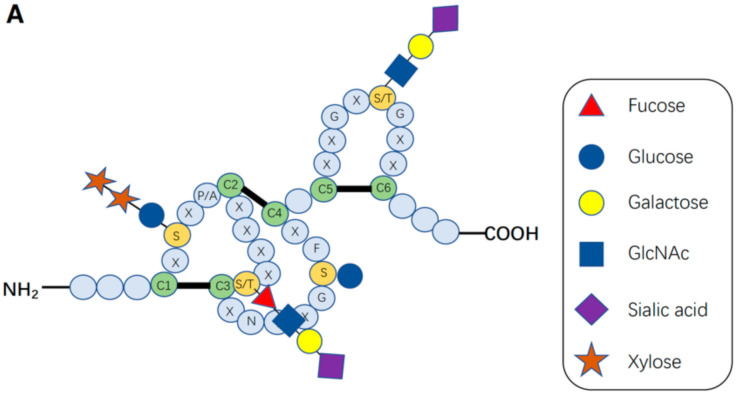
*O*-Glycosylation in the Notch EGF repeats. (**A**) *O*-Glycosylation on single EGF repeats. The EGF repeats can be modified with *O*-glycans at distinct sites. The three major types of *O*-linked glycosylation found on the EGF repeats of Notch pathway components are *O*-fucosylation, *O*-glucosylation, and *O*-*N*-acetylglucosamine (GlcNAc) modification, which exist as monosaccharides or extended forms. *O*-Fucose monosaccharides are added to a serine or threonine residue within the consensus sequence, C^2^-X-X-X-X-(S/T)-C^3^, and can be elongated with GlcNAc residues by a Fringe family GlcNAc-transferase. *O*-Glucose monosaccharides are added by protein *O*-glucosyltransferase (POGLUT)-1 to a serine residue within the consensus sequence, C^1^-X-S-X-(P/A)-C^2^, and can be further elongated to xylose-glucose-*O* disaccharides and xylose-xylose-glucose-*O* trisaccharides by glucoside α-1,3-xylosyltransferases (GXYLT) and xyloside α-1,3-xylosyltransferase-1 (XXYLT1). *O*-Glucose monosaccharides are added by POGLUT2 or POGLUT3 to a serine residue within the consensus sequence, C^3^-X-N-T-X-G-S-F-X-C^4^. *O*-GlcNAc monosaccharides are added to a serine or threonine residue within the consensus sequence, C^5^-X-X-G-X-(S/T)-G-X-X-C^6^, and can be modified with galactose and sialic acid in mammals. (**B**) Schematic of *O*-glycosylation sites on EGF repeats in the ECD of mouse NOTCH1. Data are from references [12,22,23]. Grey-colored sugar indicates that the predicted site appears to be unmodified, and white-colored sugar indicates that data for that site are not yet available. (**C**) Schematic of *O*-glycosylation sites on EGF repeats in the ECD of mouse NOTCH2. Data are from ref [12,24]. Grey-colored sugar indicates that the predicted site appears to be unmodified, while white-colored sugar indicates that data for that site are not yet available.

**Figure 3 molecules-27-01783-f003:**
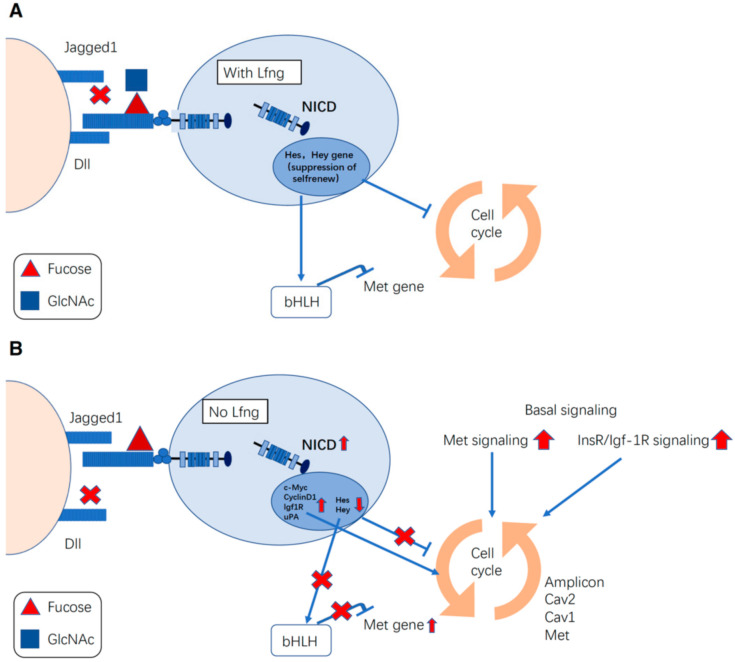
Significant involvement of lunatic Fng (LFNG) in triple-negative breast cancer (TNBC) [134]. (**A**) Activation of Notch signaling activation with Lfng. LFNG can enhance the binding between Notch receptor to DLL ligand in TNBC. (**B**) Activation of Notch signaling without Lfng in TNBC. LFNG deficiency can lead to increased JAG1-mediated Notch activation, and upregulated C-MYC, Cyclin D1, insulin-like growth factor 1 receptor (Igf-1R), and urokinase plasminogen activator (uPA) expression resulted in increasing proliferation. Meanwhile, Met/Caveolin Amplicon increased the abundance of Met and Caveolin 1/2, whose proteins can enhance signaling via IGF-1R and insulin-receptor substrate (IRS). Met and IGF-1R signaling can also stimulate proliferation.

**Table 1 molecules-27-01783-t001:** Association between glycosyltransferase related genes and cancer.

*O*-Glycosylation	Glycosyl-Transferase Gene	Cancer	Alteration	Reference
*O*-Glucosylation	*POGLUT1*	Acute myeloid leukemia	Amplification	Teng et al., 2006
	*POGLUT1*	T-cell acute lymphoblastic leukemia	Amplification	Wang et al., 2010
	*POGLUT1*	Non-small cell lung cancer	Amplification	Chammaa et al., 2018
	*GXYLT2*	Acute lymphoblastic leukemia	Upregulation	Wouters et al., 2009
	*XXYLT1*	Squamous cell carcinoma	Amplification	Yu et al., 2015
*O*-Fucosylation	*POFUT1*	Hepatocellular carcinoma	Amplification	Ma et al., 2016
	*POFUT1*	Colorectal cancer	Amplification	Chabanais et al., 2018
	*POFUT1*	Colorectal cancer	Gain-of-function mutation	Deschuyter et al., 2020
	*POFUT1*	Brain tumor	Upregulation	Kroes et al., 2007
	*POFUT1*	Squamous cell carcinoma	Upregulation	Yokota et al., 2013
	*LFNG*	Basal like Cancer	Low-expression	Xu et al., 2012
	*LFNG*	Metastatic melanoma	Absence	Del Castillo Velasco-Herrera et al., 2018
	*MFNG*	Coronal cancer	Low-expression	Lopez-Arribillaga et al., 2018
	*MFNG*	Claudin-Low Breast Cancer	Over-expression	Zhang et al., 2013
*O*-GlcNAcylation	*EOGT*	Pancreatic ductal adenocarcinoma	Upregulation	Barua et al., 2021

## Data Availability

Not applicable.

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
