# Peer review of "Significant Roles of Notch O-Glycosylation in Cancer"

_molecules, 2022, doi:10.3390/molecules27061783_

Round 1

Reviewer 1 Report

To authors:

I do not have any major points to address as the review is focused and well written.

Minor points:

  1. In page 7, lines 230-235 it is mentioned GlcNac-fucose disaccharide can be further elongated with Gal and Neu in mammals and a reference is cited. As this is clearly a glycobiology review a short mention of which are the enzymes/genes/ linkages found would be useful without having to go into the literature.

The same can be said on pg 9 lines 356-360. What are the enzymes involved? A short sentence should be enough.

  1. There are several errors of prepositions and connectors. Some make the meanings of the sentences unclear or directly senseless. I write down some examples but is a careful revision of the text is needed.

line 286 without->but not

line 345 Furthermore ->In contrast

line 444 a mutation -> the mutation

line 456 to->in (I think because if not the sentence makes no sense)

line 490 into-> from (?)

  1. Several spacing errors throughout the manuscript, especially between text and reference, sometimes found with a space sometimes without.

Author Response

To authors: I do not have any major points to address as the review is focused and well written.

We would like to thank the reviewer for her/his positive and constructive comments on our manuscript. We fully agree with the reviewer and revised the manuscript accordingly.

Minor points:

  1. In page 7, lines 230-235 it is mentioned GlcNac-fucose disaccharide can be further elongated with Gal and Neu in mammals and a reference is cited. As this is clearly a glycobiology review a short mention of which are the enzymes/genes/ linkages found would be useful without having to go into the literature.

The same can be said on pg 9 lines 356-360. What are the enzymes involved? A short sentence should be enough.

We added the information at both places.

  1. There are several errors of prepositions and connectors. Some make the meanings of the sentences unclear or directly senseless. I write down some examples but is a careful revision of the text is needed.

line 286 without->but not

line 345 Furthermore ->In contrast

line 444 a mutation -> the mutation

line 456 to->in (I think because if not the sentence makes no sense)

line 490 into-> from (?)

Fixed accordingly.

  1. Several spacing errors throughout the manuscript, especially between text and reference, sometimes found with a space sometimes without.

Done.

Reviewer 2 Report

The authors submitted a comprehensive review on the role of Notch O-glycosylation in cancer. In general, it was very well put-together. Most of the literature reviews were up-to-date however a couple of studies have been left out and the authors may want to revisit some of the recent findings to update the review paper:

  1. The functional characterization of POFUT1 variants associated with Notch in CRC (Deschuyter et al., 2020)
  2. Recent studies have also reported the expression pattern (Cheng et al., 2021) and the role of Manic Fringe in renal cell carcinoma (angiogenesis) (Cheng et al., 2022)
  3. Table 1. to be populated with more recent findings in the last two years

Author Response

The authors submitted a comprehensive review on the role of Notch O-glycosylation in cancer. In general, it was very well put-together. Most of the literature reviews were up-to-date however a couple of studies have been left out and the authors may want to revisit some of the recent findings to update the review paper:

  1. The functional characterization of POFUT1 variants associated with Notch in CRC (Deschuyter et al., 2020)
  2. Recent studies have also reported the expression pattern (Cheng et al., 2021) and the role of Manic Fringe in renal cell carcinoma (angiogenesis) (Cheng et al., 2022)
  3. Table 1. to be populated with more recent findings in the last two years

We would like to thank the reviewer for her/his positive and constructive comments on our manuscript. In particular, raising the recent literatures that were missing in the original manuscript is helpful. Now we added the three papers to the main text and Table 1 in the revised manuscript.

Regarding Table 1, we think that it would be more beneficial for readers to include all of them with the abovementioned ones rather than the last two years. Therefore, we keep the changes to just adding the papers suggested above.

Reviewer 3 Report

This is a very interesting and comprehensive review on Notch signaling and glycosylation, especially in cancer. The Figures are excellent, and the text is interesting and well written. 

Author Response

This is a very interesting and comprehensive review on Notch signaling and glycosylation, especially in cancer. The Figures are excellent, and the text is interesting and well written. 

We would like to thank the reviewer for her/his positive comments on our manuscript.